# Age-Dependent Composition and Diversity of the Gut Microbiome in Endangered Gibbon (*Nomascus hainanus*) Based on 16S rDNA Sequencing Analysis

**DOI:** 10.3390/microorganisms13061214

**Published:** 2025-05-26

**Authors:** Jieli Fan, Yanan Yin, Yanhui Liu, Yuan Chen, Wenxing Long, Chenghong Liao

**Affiliations:** 1Laboratory of Tropical Veterinary Medicine and Vector Biology, School of Life and Health Sciences, Hainan Province Key Laboratory of One Health, Collaborative Innovation Center of One Health, Hainan University, Haikou 570228, China; fanjieli0722@163.com (J.F.); yinyanan54@163.com (Y.Y.); liuyanhui913@foxmail.com (Y.L.); 2Hainan International One Health Institute, Hainan University, Haikou 570228, China; 3Wuzhishan National Long-Term Forest Ecosystem Monitoring Research Station, Hainan Key Laboratory for Sustainable Utilization of Tropical Bioresource, College of Forestry, Hainan University, Haikou 570228, China; yuanchen@hainanu.edu.cn (Y.C.); oklong@hainanu.edu.cn (W.L.); 4Institute of Hainan National Park, Haikou 570228, China

**Keywords:** Hainan gibbons, gut microbiota, age-related variation, 16S rRNA

## Abstract

The Hainan gibbon (*Nomascus hainanus*) is one of the most endangered primates globally, threatened by habitat destruction, genetic diversity loss, and ecological competition. In this study, given the critical role of the gut microbiota in host immune regulation and nutrient metabolism, we investigated the composition of and age-related variations in the gut microbiota in Hainan gibbons. Using 16S rRNA sequencing, we systematically investigated the gut microbial diversity of Hainan gibbons. We collected 41 fecal samples from Hainan Tropical Rainforest National Park, covering three age groups: juveniles (4–6 years), subadults (7–10 years), and elderly animals (≥13 years). This study found that microbiota composition changed significantly with age. Juveniles had higher microbial diversity and complexity, while subadults showed an increased abundance of *Fibrobacter* and *Prevotella* in their microbial communities, along with a Tax4Fun-predicted enrichment of functional genes related to energy metabolism, cell motility, and nervous system functions. LEfSe analysis identified statistically significant microbial taxa among different age groups, with Bacteroidota and Firmicutes being the dominant phyla across all groups with varying proportions. These results highlight the critical role of the gut microbiota in the health and adaptability of Hainan gibbons, offering insights for conservation strategies. The findings of this study are significant for understanding the changes in gut microbiota and their ecological functions across different life stages of endangered primates.

## 1. Introduction

The Hainan gibbon (*Nomascus hainanus*) is an endangered species endemic to China, belonging to the order Primates, family Hylobatidae, and genus *Nomascus* [1,2]. As one of the most critically endangered primates in the world, the Hainan gibbon is listed as “Critically Endangered” by the International Union for Conservation of Nature (IUCN) [3]. Currently, there are only 35 individuals remaining, organized into five family groups [4], and they are confined to the Hainan Tropical Rainforest National Park, particularly in the former Bawangling National Nature Reserve and the adjacent Dongbengling area [5,6].

The survival of the Hainan gibbon is primarily threatened by habitat destruction, loss of genetic diversity, and increased ecological competition. Habitat destruction is mainly driven by human activities [7,8], while the loss of genetic diversity stems from historical population bottlenecks [3,9,10]. Ecological competition has intensified due to the decline in habitat quality [7,8]. The genetic fragility of the Hainan gibbon makes it difficult for the species to adapt to rapid environmental changes. Moreover, their diet mainly consists of fleshy, juicy fruits such as berries, drupes, and collective fruits [11,12,13]. However, their specialized diet and the seasonal variability of food resources can result in nutritional deficiencies [12,14]. These factors collectively threaten the survival and reproduction of the Hainan gibbon.

The gut microbiota play essential roles in host immune regulation and nutrient metabolism, including food digestion, immune homeostasis, pathogen defense, reproduction, fat deposition, and brain development [15]. Research has shown that the diversity and function of the gut microbiota are closely linked to the host’s age, health status, and environmental interactions [16,17]. The gut microbiota begin to establish before birth and develop rapidly with age [18]. For example, young *Rhinopithecus roxellana hubeiensis* typically exhibit lower microbial diversity, but their gut microbiota become more complex and stable as they age, aiding in disease resistance and nutrient absorption [19].

This study aims to explore the diversity of gut microbial communities and their age-related changes in Hainan gibbons using high-throughput 16S rRNA gene sequencing. We seek to understand the potential impacts of these microbial changes on Hainan gibbons to provide a scientific basis for their effective management and conservation.

## 2. Materials and Methods

### 2.1. Research Location

In this study, fecal samples of Hainan gibbons were collected from the Hainan Tropical Rainforest National Park (N 18°57′–19°11′, E 109°03′–109°17′) in November 2021. Covered by tropical rainforest, the reserve has an elevation of 590–1560 m. The area exhibits a tropical monsoon climate, marked by a dry season from November to the following April and a rainy season from May through October. The average annual rainfall is 1657 mm and the average annual temperature is 21.3 °C. Eight Hainan gibbons in Group C were divided into three different age groups, including juveniles (Juv, 4–6 years old), subadults (Sub, 7–10 years old), and elderly animals (Eld, ≥13 years old).

### 2.2. Sample Collection

A total of 41 fresh fecal samples were collected from group C of Hainan gibbons in November 2021, including 7 samples from juveniles, 15 from subadults, and 19 from elderly individuals. The sampling process was conducted as follows: Fecal sampling was assisted by experienced members of the Hainan gibbon monitoring team from the Bawangling Forestry Bureau. The monitoring stations were reached in the morning before the Hainan gibbons chirp. Family groups of Hainan gibbons were located by chirping and tracked with binoculars. Individuals were identified and feces collected as soon as they were found, based on size, fur color, morning chirps, and other behaviors. Samples were only collected from individuals observed while defecating in the field to ensure that the feces were fresh. An uncontaminated mid-portion of fecal material was collected using a sterile tool and placed in 50 mL sterile centrifuge tubes, sealed, labeled, and temporarily stored in an ice box for later storage at −80 °C.

### 2.3. DNA Extraction and PCR Amplification

According to the manufacturer’s protocols, microbial DNA was extracted using HiPure Stool DNA Kits (Magen, Guangzhou, China). Nanodrop spectrophotometers (Thermo Fisher Scientific, Waltham, MA, USA) were used to verify the concentration and quality of microbial DNA. The V3-V4 regions of the bacterial 16S rRNA gene (from 341 to 806) were amplified by PCR (95 °C for 5 min, followed by 30 cycles of 95 °C for 1 min, 60 °C for 1 min, and 72 °C for 1 min and a final extension at 72 °C for 7 min) using primers 341F (5′-CCTACGGGNGGCWGCAG-3′) and 806 R (5′-GGACTACHVGGGTATCTAAT-3′). A 50 μL mixture containing 10 μL 5 × Q5@ Reaction Buffer, 10 μL 5 × Q5@ High GC Enhancer, 1.5 μL 2.5 mM dNTPs, 1.5 μL of each primer (10 μM), 0.2 μL Q5@ High-Fidelity DNA Polymerase, and 50 ng template DNA was used. The corresponding PCR reagents were purchased from New England Biolabs, Ipswich, MA, USA.

### 2.4. 16S rRNA Gene Sequencing

Amplicons were extracted from 2% agarose gels and purified using the AxyPrep DNA gel extraction kit (Axygen Biosciences, Union City, CA, USA) according to the manufacturer’s instructions and quantified using the ABI StepOnePlus Real-Time PCR System (Life Technologies, Foster City, CA, USA). The purified amplicons were then pooled in an equimolar fashion and subjected to paired-end sequencing (PE250) on an Illumina platform according to standard protocols.

### 2.5. Processing of 16S rRNA Gene Sequencing Data

After sequencing, the raw reads were filtered using FASTP [20] (version 0.18.0) to remove reads containing more than 10% unresolved nucleotides (N) and reads containing less than 80% bases with a Q value of >20. Paired-end clean reads were merged as raw tags using FLASH [21] (v 1.2.11) with a minimum overlap of 10 bp and mismatch error rates of 2%. Noisy sequences of raw tags were filtered by the QIIME [22] (V1.9.1) pipeline under specific filtering conditions to obtain high-quality clean tags. The clean tags were clustered into operational taxonomic units (OTUs) of ≥97% similarity using the UPARSE [23] (version 9.2.64) pipeline. All chimeric tags were removed using the UCHIME algorithm [24] and effective tags were finally obtained for further analysis. The tag sequence with the highest abundance was selected as the representative sequence within each cluster. The representative sequences were classified into organisms by a naive Bayesian model using the RDP classifier [25] (version 2.2) based on the SILVA [26] database (version 132), with the confidence threshold values ranging from 0.8 to 1. All procedures were performed by Gene Denovo Biotechnology (Guangzhou, China). Bioinformatic analysis was performed via Omicsmart, a Dynamic Real-Time Interactive Online Platform for Data Analysis (http://www.omicsmart.com).

## 3. Results

### 3.1. Analysis of 16S rRNA Gene Sequencing Data

To thoroughly investigate the gut microbiome characteristics of Hainan gibbons across different age stages, we sequenced the 16S rRNA V3-V4 hypervariable region in fecal samples from wild Hainan gibbons, thereby comprehensively characterizing their gut microbiota. The aim was to identify microbial taxa with increased abundance and key driver microbes in different age groups, as well as to explore age-related microbial phenotypes and functional features. A total of 41 fecal samples from Hainan gibbons were analyzed, with the initial sequencing generating 4,969,594 raw tags. After the removal of low-quality sequences and chimeras, 4,932,719 high-quality tags were obtained, averaging approximately 120,310 ± 14,097 clean tags per sample and 96,004 ± 9677 effective tags, with an effective ratio exceeding 75.72%. Within these effective tags, the maximum and minimum lengths were 478 ± 1 and 203 ± 4, respectively. Based on 97% sequence similarity, these effective tags were classified into 16,914 operational taxonomic units (OTUs), averaging 413 OTUs per sample. As part of the genome assembly quality assessment, the average N50 value exceeded 461, indicating a high degree of completeness and good quality of the sample genome assembly, thus providing a reliable basis for subsequent analyses (Appendix A). All samples’ Shannon rarefaction curves showed a plateau stage, indicating adequate sampling of 16S rRNA se quences for all the samples (Appendix A).

### 3.2. Dominant Bacterial Community Across Age Groups

A taxonomic summary of the microbial components in all samples identified 20 bacterial phyla, 32 classes, 88 orders, 120 families, and 173 genera. We presented the top 10 bacterial phyla in terms of relative abundance across different age groups using stacked bar charts, with these phyla accounting for over 98.86% of the total. In the three age groups, Bacteroidota, Firmicutes, and Fibrobacterota were the predominant phyla, with their abundance in the juveniles (Juv), subadults (Sub), and elderly individuals (Eld) being as follows: Bacteroidota (Juv: 50.45% ± 7.47%; Sub: 42.91% ± 4.58%; Eld: 43.37% ± 3.99%), Firmicutes (Juv: 26.72% ± 8.52%; Sub: 30.73% ± 3.37%; Eld: 32.36% ± 4.77%), and Fibrobacterota (Juv: 11.57% ± 10.31%; Sub: 17.54% ± 4.71%; Eld: 16.82% ± 3.19%). The ratio of Firmicutes to Bacteroidota (F/B) in the Juv, Sub, and Eld groups was 55.19% ± 21.46%, 72.65% ± 12.18%, and 75.86% ± 15.84%, respectively (Figure 1A). Additionally, our analysis showed that 62.21% of the sequences could be annotated at the genus level. A heatmap revealed the top twenty bacterial genera in terms of relative abundance across different samples, including six genera from Bacteroidota (*Prevotella*, *Alloprevotella*, *Prevotellaceae_UCG-001*, *Prevotellaceae_NK3B31_group*, *Rikenellaceae_RC9_gut_group*, *Bacteroides*), eight from Firmicutes (*Phascolarctobacterium*, *Lachnospiraceae_NK3A20_group*, *Asteroleplasma*, *UCG-004*, *Lachnospira*, *Solobacterium*, *Lachnospiraceae_UCG-004*, *Colidextribacter*), and *Fibrobacter* from Fibrobacterota. Among the age groups, the dominant genera were *Fibrobacter* (Juv: 11.57% ± 10.31%; Sub: 17.54% ± 4.71%; Eld: 16.82% ± 3.19%), followed by *Prevotella* (Juv: 14.43% ± 1.77%; Sub: 14.65% ± 2.61%; Eld: 14.84% ± 1.95%), *Phascolarctobacterium* (Juv: 5.80% ± 4.37%; Sub: 5.56% ± 3.04%; Eld: 6.58% ± 1.91%), and *Lachnospiraceae_NK3A20_group* (Juv: 5.21% ± 3.36%; Sub: 4.45% ± 1.87%; Eld: 6.28% ± 1.89%) (Figure 1B).

### 3.3. Comparative Analysis of Microbial Diversity Across Age Groups

By employing the Tukey HDS test method, we created a box plot illustrating the α diversity Shannon index (Figure 2A). This plot displayed the Shannon indices for the three different age groups (Juv: median = 5.14; Sub: median = 4.85; Eld: median = 4.88). Statistical analysis indicated that although the α diversity index was slightly higher in the juvenile group, there was no significant variation in α diversity across the three age groups. Furthermore, the α diversity indices for the subadult and elderly groups exhibited lower dispersion and higher stability. In the analysis of β diversity, we utilized non-metric multidimensional scaling (NMDS) analysis based on Bray–Curtis distance to assess community structure differences at the genus level (Figure 2B). The stress value of this NMDS analysis was 0.096 (stress < 0.1), indicating that the model was reliable and accurate. The analysis revealed that the microbial community structures of the subadult and elderly groups were quite similar. Furthermore, ANOSIM analysis at the OTU level further revealed significant differences in microbial community structure between the different age groups. A comprehensive comparison among the Juv, Sub, and Eld groups (R = 0.3779, *p* = 0.001) demonstrated statistically significant differences in community structure. More detailed pairwise comparisons indicated substantial differences in microbial community structure between the Juv and Sub groups (R = 0.7494, *p* = 0.001), as well as between the Juv and Eld groups (R = 0.8202, *p* = 0.001), highlighting a clear separation in microbial composition among these age stages. However, the difference in microbial community structure between the Sub and Eld groups (R = 0.045, *p* = 0.144) was not significant, suggesting a closer similarity in microbial composition between these two age groups. These findings emphasize the dynamic changes in microbial community structure across different life stages, particularly underscoring the significant differences between the juvenile stage and other age groups (Figure 2C).

### 3.4. Comparative Analysis of Species Differentiation Across Age Groups

Linear Discriminant Analysis Effect Size (LEfSe) identified microbiota with statistical dominance in the three age groups of Hainan gibbons. The LEfSe analysis recognized 35 taxa with significant differences among the Juv, Sub, and Eld groups, revealing 21, 12, and 2 distinct taxa at different taxonomic levels with LDA scores greater than 3.0 (Figure 3A). Specifically, in the Juv group, dominant taxa included Bacteroidota, *Atopobium*, *Demequina*, Proteobacteria, *Sutterella*, *Motilimonas*, *Tyzzerella*, and *ZOR0006*. The Sub group was characterized by *Rikenellaceae_RC9_gut_group*, *Alloprevotella*, *Bifidobacterium*, *Lachnospiraceae_UCG_004*, and *Erysipelatoclostridiaceae.UCG_004*. For the Eld group, Veillonellales_Selenomonadales and Veillonellaceae were the dominant taxa (Figure 3B).

### 3.5. Functional Prediction Analysis of Gut Microbiome Across Age Groups

In this study, we utilized the Tax4Fun analytical approach to generate a bar chart of the KEGG pathways (Figure 4A). This chart was designed to visually depict the overall relative abundance distribution of all samples at KEGG Level B. At Level A, based on relative abundance, the predominant pathways were identified as Metabolism, Environmental Information Processing, Genetic Information Processing, Cellular Processes, Human Diseases, and Organismal Systems. On the more detailed Level B, carbohydrate metabolism was observed to have the highest relative abundance, followed by membrane transport and amino acid metabolism. In the analysis of the functional heatmap, we observed significant differences in microbial function at the KEGG Level B tier across different age groups (Figure 4B).

To thoroughly investigate the differences in gut microbial functions among the various age groups, our study employed Welch’s *t*-test (with a significance threshold set at *p* < 0.05) to compare the relative abundance in KEGG Level B pathways across these groups. This statistical analysis revealed significant disparities in metabolic pathways among different age cohorts. Specifically, compared to the Juv, both the Sub and Eld groups exhibited a marked increase in relative abundance in the energy metabolism, cell motility, and nervous system pathways, while showing a significant decrease in the lipid metabolism and transport and catabolism pathways. Additionally, compared between the Juv and Eld groups, the Juv group demonstrated a higher relative abundance in the metabolism of other amino acids and the endocrine system pathways compared to the Eld group but lower in the metabolism of cofactors and vitamins, as well as in endocrine and metabolic disease pathways (Figure 5A,B). The differences between the Sub and Eld groups were relatively minor, yet there was a slight increase in relative abundance in the cell motility pathway in the Sub group (Figure 5C).

## 4. Discussion

This study utilized 16S rRNA gene sequencing to investigate gut microbial diversity and its age-related dynamics in Hainan gibbons. The results showed that Bacteroidota and Firmicutes were the predominant bacterial phyla across all age groups (Figure 1A), aligning with findings in humans [27,28] and other non-human primates [29,30,31]. These bacterial groups play essential roles in gut health and host nutrient metabolism. The Firmicutes/Bacteroidota (F/B) ratio is considered an indicator of gut microbiota maturity and development [32]. A high proportion of Bacteroidota in juveniles is particularly beneficial for the maturation of the immune system, enhancing host immunity [33] and maintaining gut balance [34]. Variations in the F/B ratio are closely associated with health status and diseases such as obesity [35], emphasizing the significance of gut microbial diversity in primate physiological development and disease prevention. At the genus level, *Fibrobacter*, *Prevotella*, and *Phascolarctobacterium* showed high relative abundance across different age groups. *Fibrobacter*, typically found in the rumen of ruminants, can degrade cellulose and hemicellulose, thus enhancing host nutrient absorption [36,37]. *Prevotella*, associated with plant-rich diets [38], can digest non-cellulosic polysaccharides, pectin, and soluble sugars as energy sources [39,40]. *Phascolarctobacterium* is known for its role in the production of butyrate, a crucial energy source for colonic epithelial cells with significant anti-inflammatory effects. *Phascolarctobacterium* promotes gut health and prevents related diseases by effectively producing butyrate through the fermentation of cellulose and other indigestible carbohydrates [41,42].

Microbial diversity analysis revealed that the Juv group had higher gut microbial diversity compared to the Sub and Eld groups (Figure 2). This pattern is consistent with findings in other primates and humans. Studies by Yatsunenko et al. [43] and Sang et al. [44] indicated that young individuals have more diverse gut microbiota due to their adaptation to variable nutritional and environmental conditions. Similarly, Reese et al. [45] showed that young wild chimpanzees have higher microbial diversity than adults, attributed to their extensive food sources, immature immune systems, and rich environmental exposure. These findings underscore the biological and environmental adaptability roles of gut microbial diversity during primate development. Our study also demonstrated that the gut microbial community structures of subadult and adult groups exhibited high spatial similarity, with no significant difference in β-diversity between these age groups as indicated by NMDS analysis (Figure 2B). The stability of this community structure is likely related to the maturity of the host and its relatively stable diet and lifestyle. For instance, Faith et al. [46] found that in adults and subadults, the gut microbial community shows significant stability under consistent dietary and lifestyle conditions. These findings highlight the importance of stable dietary habits in mature hosts for gut microbial diversity and community structure stability, revealing the biological basis of microbial community stability with age.

LefSe analysis indicated significant differences in microbial groups enriched in the different age groups of Hainan gibbons. In the juvenile group, Bacteroidota predominate and support rapid growth and immune system maturation by their association with carbohydrate metabolism and mucosal immunity. Bacteroidota bacteria break down complex carbohydrates to produce short-chain fatty acids (SCFAs), which provide energy to the host, regulate the gut environment, and enhance mucosal immunity by promoting IgA responses [47,48,49]. In the subadult group, *Alloprevotella* dominate and play a key role in degrading cellulose and polysaccharides, producing short-chain fatty acids such as propionate to meet the energy needs and help regulate metabolism and reduce inflammation [50,51]. *Alloprevotella* also synthesize B vitamins, supporting energy metabolism and nervous system function, and maintaining gut health [52]. In the elder group, Veillonellaceae were significantly enriched; these bacteria are associated with butyrate production, essential for maintaining gut health and reducing inflammation [53]. The higher relative abundance of Veillonellaceae in elderly individuals helps alleviate inflammation and maintain gut barrier function. Research has also shown that butyrate plays a key role in maintaining intestinal epithelial cell integrity and regulating immune responses [54]. These findings are consistent with studies in other primates and humans. Odamaki et al. [16] pointed out that gut microbial composition changes with age from newborns to centenarians. Reese et al. [45] observed higher gut microbial diversity in young wild chimpanzees compared to adults, aligning with our findings. These studies indicate that age-related changes in gut microbiota highlight the crucial role of the microbiome in supporting host health and physiological functions at different life stages.

Functional prediction analysis showed significant differences in gut microbial functions across different age groups, reflecting their physiological needs and health status. The Juv group exhibited relatively higher predicted abundances of microbial pathways associated with amino acid metabolism, endocrine interactions, and lipid metabolism, supporting rapid growth and immune system development. Such findings align with related studies indicating that gut microbiota play a key role in modulating endocrine functions, which in turn regulate host growth and immune responses [55]. The Sub and Eld groups showed significant advantages in energy metabolism, metabolism of cofactors and vitamins, and cell motility, reflecting their high energy demands and the metabolic needs of a high-fiber diet. Studies have pointed out that as individuals mature, the energy metabolism capacity of the gut microbiome significantly increases, supporting high energy demands [40]. These results are consistent with recent studies on the gut microbiome functions of Hainan gibbons [29], indicating that the functional roles of the gut microbiota significantly influence the physiological needs and health status of hosts at different ages.

## 5. Conclusions

This study, through 16S rRNA gene sequencing, revealed significant age-related differences in the gut microbiota of Hainan gibbons. Juveniles exhibited higher microbial diversity, while adults had a greater abundance of genes related to energy metabolism, cell motility, and nervous system functions. Dominant phyla like Bacteroidota and Firmicutes showed significant variations across age groups. These findings underscore the critical role of the gut microbiota in the health and adaptability of Hainan gibbons, providing a scientific basis for conservation strategies. Future research should further explore the mechanisms of these microbial changes and their long-term impacts, advancing primate conservation science.

## Figures and Tables

**Figure 1 microorganisms-13-01214-f001:**
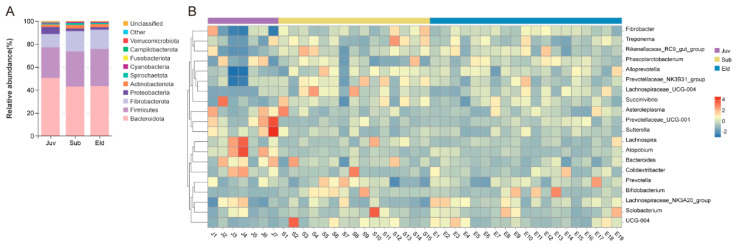
Microbial composition across age groups. (**A**) Stacked bar chart showing the top 10 bacterial phyla by relative abundance in the three age groups. (**B**) Heatmap of the top 20 bacterial genera by relative abundance across age groups.

**Figure 2 microorganisms-13-01214-f002:**
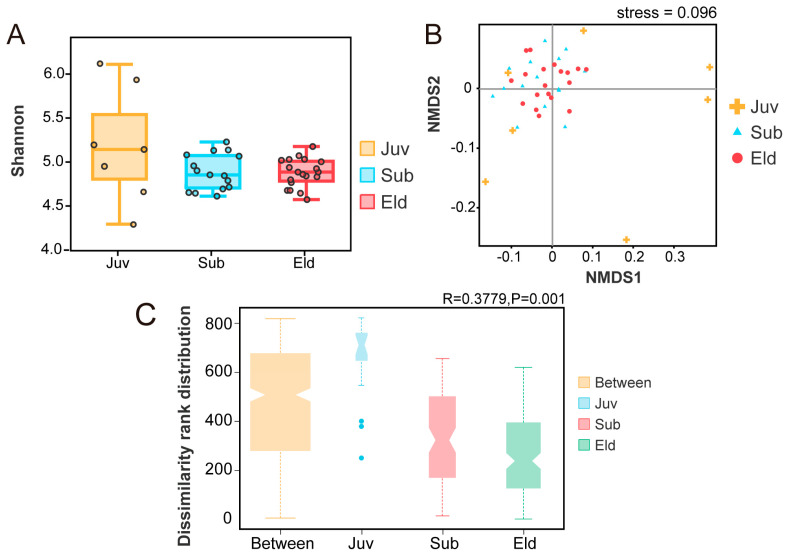
Analysis of α and β diversity across different age groups. (**A**) Boxplot showing Shannon diversity indices for α diversity across three age groups. (**B**) NMDS analysis depicting β diversity at the genus level based on Bray–Curtis distance, stress = 0.096. (**C**) ANOSIM analysis at the OTU level, with the vertical axis indicating distance rankings.

**Figure 3 microorganisms-13-01214-f003:**
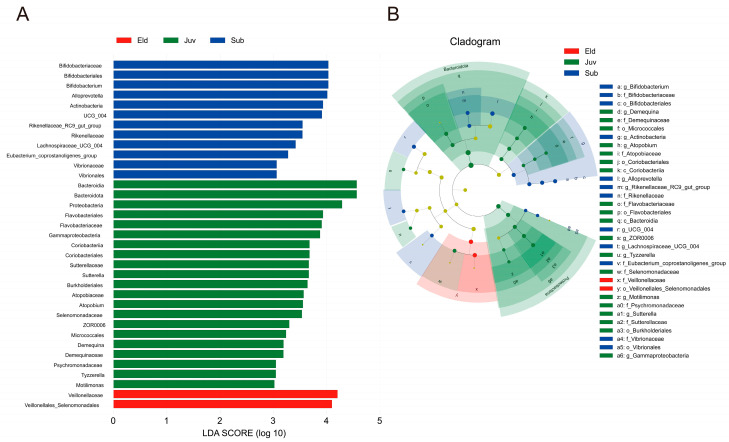
LefSe analysis. (**A**) Histogram of LDA values, showing biomarkers with LDA scores ≥3.0 and statistically significant differences (*p* < 0.05). (**B**) Phylogenetic cladogram based on LefSe analysis, displaying taxonomic levels from phylum to genus.

**Figure 4 microorganisms-13-01214-f004:**
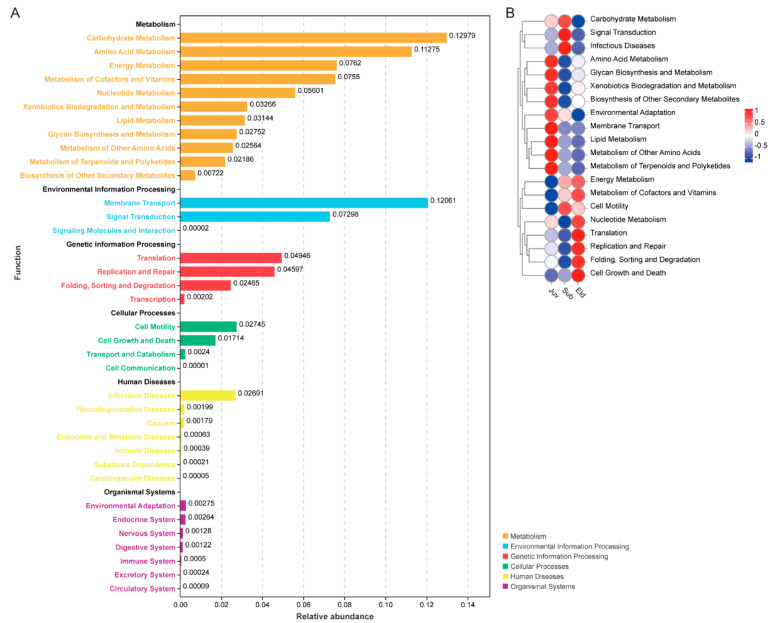
KEGG pathway enrichment analyzed by Tax4Fun. (**A**) The bar chart displays the relative abundance of KEGG Level B pathways across the entire sample set, with different colors distinguishing the Level A categories. (**B**) Functional abundance heatmap based on Tax4Fun analysis, showing differences in the relative abundance of Level B pathways across different groups.

**Figure 5 microorganisms-13-01214-f005:**
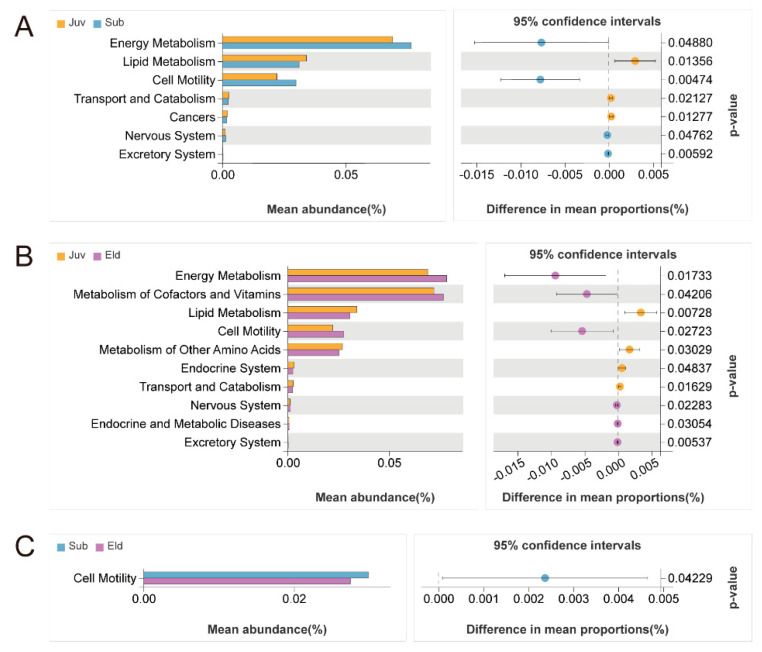
Welch’s *t*-test analysis of functional differences in KEGG level B pathways across different age groups (*p* < 0.05). (**A**) Comparison of pathway abundance between the juvenile group Juv and the subadult group Sub; (**B**) comparison between the juvenile group Juv and the elderly group Eld; (**C**) comparison between the subadult group Sub and the elderly group Eld.

## Data Availability

The 16S rRNA sequencing raw data for *Nomascus hainanus* are available at NCBI BioProject: PRJNA1153314. The link is https://dataview.ncbi.nlm.nih.gov/object/PRJNA1153314 (accessed on 30 August 2024).

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
