# Peer review of "Age-Dependent Composition and Diversity of the Gut Microbiome in Endangered Gibbon (Nomascus hainanus) Based on 16S rDNA Sequencing Analysis"

_microorganisms, 2025, doi:10.3390/microorganisms13061214_

Round 1
Reviewer 1 Report
Comments and Suggestions for Authors
Dear authors,
There are some aspects that require your attention, such as :
C1 - References 11 and 12 cannot be evaluated as they are only available in Chinese ( line 50, 52 )
C2 - Line 155,156 - ratio F/B - Could you be more clear regarding these values in relation with the Figure 2A? The ratio is B/F acc. Figure 2 A
C3 - ref. 44 is in relation only with the human gut microbiota
C4 - line 286 - the references 22 represents the study of Bokerlich et al. and there isn't a relation with the statement
C5 - line 301 - the reference 51 is in relation with human beings and vervets ( fam. Cercopithecidae ). The gibbons. mention in your statament, belongs to the family Hylobatidae.
C6 - reference 51, line 301 - Please check to see how reference 51 relates to your statement.
C7 - the reference 52, line 302 is not the study of Chivers et al.
C8 - the reference 53 don't include information in relation with Veillonellaceae
C9 - Please check the the technical writing requirements of the journal. for the list of bibliographic references
Reviewer 2 Report
Comments and Suggestions for Authors
The work is designed and executed as correctly as possible. The group is not too large and drawing conclusions based on several samples is burdened with a fairly high risk of error. I understand that the number of individuals available could be a problem here. Below are detailed comments on the work:
The chapter "Research Location" provides different numbers of age groups than the chapter "sample collection".
There is a typo in the methods FLSAH - FLASH.
Figure 1. I think it can be moved to the supplement.
Figure 3 is difficult to read due to its small size and images A and C have a different style of box plots. For consistency, it would be worth standardizing this. I don't know if the marked elliptical ranges are necessary on the NMDS graph. One group is so diverse that everything fits into it.
Let the authors check the use of italics for Latin names in the text.
